# Salicylic Aldehyde and Its Potential Use in Semiochemical-Based Pest Control Strategies Against *Trypophloeus binodulus*

**DOI:** 10.3390/insects15120964

**Published:** 2024-12-04

**Authors:** Antonio Ortiz, Lucía Ruiz-Martos, Andy Bruno, Carmen Vega-Valdés, Eva Díez-Presa, Lucía Delgado-Salán, Dana Mínguez-Bermejo, Pedro A. Casquero, Álvaro Rodríguez-González

**Affiliations:** 1Inorganic and Organic Department, EPS Linares, University of Jaén, 23700 Linares, Spain; lrm00050@red.ujaen.es (L.R.-M.); ajbd0001@red.ujaen.es (A.B.); 2Grupo Universitario de Investigación en Ingeniería y Agricultura Sostenible (GUIIAS), Instituto de Medio Ambiente Recursos Naturales y Biodiversidad, Escuela de Ingeniería Agraria y Forestal, Universidad de León, Avenida de Portugal 41, 24009 León, Spaindmingb00@estudiantes.unileon.es (D.M.-B.);

**Keywords:** *Trypophloeus binodulus*, Scolytidae, *Populus*, Salicaceae, salicylaldehyde, kairomone

## Abstract

Plant-derived semiochemicals have attracted considerable attention in recent years as promising attractants for the management of forest pests, mainly due to their safety and low cost. The poplar bark beetle, *Trypophloeus binodulus* (Ratzeburg, 1837) (Coleoptera: Scolytidae), is a serious pest of poplar trees (Salicaceae: *Populus*), resulting in significant economic and ecological losses across northern Spain. We tested the hypothesis that *Trypophloeus binodulus* is attracted to compounds released by its host-plant genus. This study tested adults of *T. binodulus* using electroantennography and by monitoring their populations in baited traps in northern Spain. The volatiles released by the leaves and bark of clones revealed high emissions of different chemical compounds, including benzenoids, sesquiterpenes, monoterpenes, fatty acids, and alkanes. The benzenoid salicylaldehyde dominated these collections and was more abundant in aerations of poplar leaves than in other odor sources. Field tests showed that traps baited with salicylaldehyde and ethanol captured more adults than all other treatments. These findings highlight the potential of salicylaldehyde for managing *T. binodulus* and developing environmentally friendly pest control strategies.

## 1. Introduction

Plants produce various volatile organic compounds (VOCs) in different quantities and ratios [1]. These substances play a key role in plant defense as well as in the detection and acceptance of plant hosts by herbivorous insects [2,3,4]. Kairomones are chemical signals that mediate interspecific interactions that are beneficial to organisms that detect these cues, and they play a relevant role for bark beetles [5]. Woodboring insects are frequently attracted to volatile compounds associated with hosts [6], especially those in a stressed physiological state, and to pheromones produced by sexually mature conspecifics [7,8,9]. Many insect species in the family Scolytidae (Coleoptera) are among the most significant and economically important pest insects in the northern hemisphere, and they are capable of killing even healthy trees in natural and commercial forests [10,11] as a result of mass colonization [12]. Bark beetles, in particular, utilize semiochemicals when locating and colonizing host trees, mating, or interacting with mutualistic species [2,13,14,15].

Notably, when trees are stressed by drought, disease, or wildfire, their defenses are diminished, making them more vulnerable to beetle colonization. Bark beetles [12] can benefit from aggregation on their host plants, as feeding in groups helps to overcome host defense mechanisms, increases the probability of finding a mating partner, and reduces predation pressure from natural enemies [16,17]. During the recognition, selection, and colonization process [18], bark beetles move around within a given forest ecosystem and precisely select specific host trees that greatly reduce the threat they might pose to them [19]. Identifying an optimal host facilitates energy allocation for reproduction and attenuates a foraging individual’s exposure to hazards such as predation, desiccation, or inclement weather. Bark beetles from the subfamily Scolytinae are among the most important pests in forests of Europe [20]. *Trypophloeus* spp. (Coleoptera: Curculionidae: Scolytinae) is a genus that includes, to date, according to the EPPO database, the species *T. binodulus, T. asperatus*, *T. granulatus*, *T. klimeschi*, *T. populi*, and *T. rybinskii*, as well as others not classified (spp.), which can generally be devastating pests for hybrid or natural poplar crops [20,21]. These beetles invade shoots first and then gradually spread to the main stem. The eggs are deposited in the phloem under the tree’s bark, where the larvae feed until they pupate and emerge as adults to find a new host tree.

There are usually two generations per year, with mature larvae overwintering. First-generation adults emerge from late May to late June, with second-generation adults emerging in August [22,23]. Adult beetles colonize the branches and stems of mature trees, boring into the bark to mate. In response to *Trypophloeus binodulus* colonization attempts by *Trypophloeus*, the tree produces a resinous gum to push out colonizing beetles as they chew through the bark. This resin causes a dark-colored stain in the wood, which is eventually incorporated into the growth ring of the tree, leading to a defect referred to as a “gum spot” [24,25].

It has already been shown in the work carried out by Lombardero in Galicia [26] that the genus *Trypophloeus*, and specially *Trypophloeus binodulus*, which is the focus of this study, can be associated with poplar plantations. *T. binodulus* exhibits a strong specificity for *Populus alba* var. *pyramidalis*, and the extent of the lesion differs with the tree’s age [27]. Host specificity suggests that some host-specific volatiles attract the pest [23]. Widespread outbreaks of this pest have occurred in the past, causing huge economic and ecological losses, as is the case of the forest refuge in northwest China [22]. In this case, the affected species was *P. alba* var. *pyramidalis*, which was used as an afforestation tree for greening and shelterbelts in the North China Plain, especially in the Three Norths Area, hence providing a sort of continuous corridor for the spread of *Trypophloeus* from the northwest to eastern China [22]. A similar situation is feared in the Castilla y León region (northwest Spain). In the poplar (*Populus nigra*) groves of Castilla y León (Northern Spain), it has also been seen that among a great variety of clones, *Trypophloeus binodulus* only attacks clone USA 184-411 [28]. This fact is important, since this clone predominates in the poplar groves of the province of Castilla y León, which supplies 60% of this type of wood to the Spanish industry.

Recognition of plant hosts by *Trypophloeus* spp. may occur through species-specific compounds or through mixtures of such compounds [29,30]. Although there are in-depth studies of various species, the semiochemistry of *Trypophloeus* spp. has not been extensively investigated, with the exception of *Trypophloeus klimeschi*, for which there are preliminary studies of kairomones [22,23,27]. In those studies [22,23,27], the researchers analyzed the main volatile compounds of the host *P. alba* var. *pyramidalis* under different physiological characteristics, as well as compounds from three other non-hosts (the hybrid *Populus dakuanensis* Hsu, *P. alba* Linnaeus, and *Populus tomentosa* Carrière). Extracts were obtained from leaves and bark separately by solid-phase microextraction (SPME) and thermal desorption. The extract samples were analyzed using gas chromatography and mass spectrometry (GC-MS). The authors also described how *T. klimeschi* induced varying levels of injury depending on the age of *P. alba* var. *pyramidalis*. The comparative analysis of the different physiological states revealed that levels of some compounds were higher in susceptible species of *P. alba* var. *pyramidalis*, and other compounds were exclusive to this host.

Based on these findings, it was decided that the volatile profiles of two poplar clones, USA 184-411 and I-214, should be evaluated to identify compounds that could mediate the identification of the host by *Trypophloeus binodulus*, and thus be able to provide candidates for semiochemicals for monitoring and control through massive capture in integrated pest management programs.

## 2. Materials and Methods

### 2.1. Insects, Traps, and Lures’ Preparation

All *Trypophloeus binodulus* spp. adults used were field-collected in the Leon province (NorthSpain) using Escolitraps^®^ (Econex, Murcia, Spain, https://www.e-econex.net/es/trampas-para-insectos/escolitrap-478.html, accessed on 12 October 2024) loaded with ethanol. These traps consist of two black paddles embedded in a black funnel, which directs the insects into a high-capacity collector. The top of the trap is fitted with a green lid with a hanger. Upon collection, the adults were sexed by visually inspecting the genital regions, transferred to 10 mL plastic cages (1 insect/cage), and fed with poplar leaves. All the beetles were reared under conditions of 25 ± 0.5 °C, 70% ± 5% relative humidity, and a 16:8 h light/dark photoperiod with 600 lx light intensity. All field-collected beetles were used within 24 h of their collection.

Ethanol lures were prepared by placing 1 mL of absolute ethanol (99.97%) on absorbent cotton and sealing it inside low-density polyethylene (LDPE) pressure-sealed bags (95 mm × 60 mm × 50 μm).

Field test stock solutions of kairomone lures were prepared by mixing *n*-hexane solutions of salicylaldehyde in a polyethylene vial. The lures were dispensed from 1.5 mL LDPE vials (31 mm long × 7 mm diameter; Semiotrap S.L., Linares, Spain). The vials were filled with 100 mg of salicylaldehyde (SA) or methyl benzoate (MB) and then heat-sealed with an aluminum film. The vials were packed in aluminum foil bags and stored at −21 °C in a freezer until further use.

### 2.2. Chemicals

Authentic chemical standards of isobutanol (95%), (*E*)-3-hexenol (96%), (*E*)-2-hexenal (>95%), (*Z*)-3-hexenol (96%), benzaldehyde (98%), limonene (>98%), 1,8-cineole (>95%), linalool (>95%), salicylaldehyde (98%), benzyl acetate (99%), catechol (>98%), salicylic alcohol (>97%), methyl salicylate (97%), methyl benzoate (>98%), α-Copaene (>90%), α Caryophyllene (90%), *trans*-(*β*)-Caryophyllene (80%), eugenol (98%), oleic acid (>80%), linoleic acid (>70%), stearic acid (95%), and methyl benzoate (>99%) were obtained from Sigma-Aldrich (Madrid, Spain). Nonanal (95%), methyl salicylate (98%), 2-phenylethanol (>95%) α-Caryophyllene oxide (90%), *trans*-β-Farnesene (90%), β-Bisabolene (>90%), hexadecanoic acid (>95%), were purchased from Fluka Ltd. (Lancashire, UK). (*E*)-cinnamyl alcohol (>98%), (*RS*)-linalool (97%), trans-nerolidol (>90%), ocimene (>95%), guaiol (>96%), geraniol (95%), β-Guaiene (>90%), and trans-β-ionone (96%) were purchased from Acros Organics (Geel, Belgium).

### 2.3. Volatiles Collection

We used SPME fibers (SPME Fiber Assembly, Supelco Co., Bellefonte, PA, USA) mounted in standard holders (Supelco, Sigma Aldrich, St. Louis, MO, USA) to collect the emitted VOCs from the leaves and bark of a *P. nigra* plantation located in Villasabariego (León province, Spain). The fibers were 1 cm long and consisted of fused silica support and a 65 μm thick PDMS/DVB (Polydimethylsiloxane/Divinylbenzene) coating. Before sampling, the SPME fibers were conditioned for 30 min at 250 °C in a gas chromatograph injection port. Samples of plant material (15–20 g of leaves or bark) were placed in separate 100 mL round bottom flasks. Prior to sampling volatiles, the flask was rinsed with hexane and dried in an oven for 10 min. The mouth (29/32 ID) of the flask was sealed with a rubber septum (30.71 mm ID, Saint-Gobain, France). For each extraction, a conditioned solid-phase microextraction fiber was then inserted through the septum and it was exposed to the headspace of the sample under room temperature (25 °C) conditions. Sampling times of 1 to 4 h were tested; 1 h was found to be sufficient to obtain good and reproducible results in the subsequent assays and was therefore selected as the optimal SPME extraction time. The test samples were run in triplicates, and final observations were presented as mean ± standard deviation.

### 2.4. Chemical Analyses

Analyses were carried out using a Thermo FOCUS GC coupled to a Thermo DSQ-II quadrupole mass spectrometer (Thermo Fisher Scientific, Waltham, MA, USA) with electron impact ionization. The GC was equipped with a DB5 column (30 m × 0.25 mm × 0.25 μm; J&W Scientific, Folsom, CA, USA) using helium as the carrier gas (1.2 mL min^−1^). The collected volatiles, adsorbed on SPME fibers, were thermally desorbed into a GC injector port at 250 °C for 3 min in splitless mode before beginning temperature programming. The column oven was maintained at 60 °C for 1 min, increased by 5 °C min^−1^ to 250 °C and held for 15 min. The transferline temperature was 280 °C. The ion source operated at a constant temperature of 200 °C with an ionization energy of 70 eV. Scanning covered the entire available range (50–500 u). The compounds were identified by comparing retention indexes (RIs) and mass spectra with the National Institute of Standards and Technology (NIST) or Wiley 275L libraries and matching their retention time and/or mass spectra to those of authentic standards. The column GC program is the one used by Adams R.P. [31] in his study to confirm the elution order in the same column.

### 2.5. Electroantennography

The electroantennography (EAG) technique was used to assess the antennal selectivity and sensitivity of adult *Trypophloeus binodulus* insect to the selected nine VOCs. All the beetles were used the next day after collecting them in the field. The EAG bioassays were performed using a Syntech system (Syntech Laboratories, Hilversum, The Netherlands). The head of each *T. binodulus* beetle was carefully excised and mounted between two metal electrodes on the antenna holder, using conductive Spectra 360 Electrode Gel (15-60, Parker Laboratories, Fairfield, NJ, USA). The head was placed on one electrode and the tip of an antenna, which is still attached to the head, on the other. The head/antennal preparation was placed under a constant stream of humidified air (flow of 500 mL min^−1^). Experimental insects were used only once.

The stimulus delivery system employed a piece of filter paper (1 × 1 cm^2^) in a disposable Pasteur pipette cartridge. A 1 μL aliquot of standard solutions of each test compound (at 10 mg mL^−1^ in HPLC grade hexane) was applied to the filter paper strips, with the solvent allowed to evaporate for 10 s before the strip was placed into the cartridge. Test stimulations were carried out by applying puffs of air (200 mL min^−1^) for 2 s using a CS-01 stimulus controller (Syntech) through the pipette. A disposable syringe blew vapor stimuli for 1 s into a constant stream of charcoal-filtered humidified air (500 mL min^−1^) flowing in a metal delivery tube (i.d. 8 mm) with the outlet positioned approximately 1 cm from the antenna holder. A control (1 μL hexane) stimulus was applied at the beginning of the experiment, and after each group of test stimuli. Puffs of the tested stimuli were applied at 1 min intervals on each dissected antenna, and the order of presentation of the test stimuli was randomized among replicates. The EAG responses were evaluated by measuring the maximum amplitude of polarity (mV) elicited by a stimulus. The absolute EAG response (mV) to each stimulus was divided by the response to the nearest control (hexane) to compensate for the solvent [32]. The response to hexane was considered a negative control, and the EAG responses were reported as relative responses to hexane.

### 2.6. Field Trapping

This experiment was carried out over a period of five months (from May to September) in 2022 in two uniformly planted poplar plantations located near the towns of Villarabariego (42°30′49.05″ N, 5°23′25.11″ W) and Villoria de Órbigo (42°25′7.15″ N, 5°52′50.33″ W). Both villages belong to the province of León, Castilla y León, Spain. The mean age of the poplars was 12 years, the mean stand density was 5 × 5 m, and the mean diameter was 30–45 cm. Soil type in the poplar stands was sandy–silty till. The test plots were surrounded by other plots of poplars.

The traps were deployed in the poplar plantations in Villarabariego and Villoria de Órbigo on 18 May 2022. The traps were divided into two blocks (one in the I-214 clone and one in the USA 184-411 clone), each with an area of 0.4 ha (40 m length × 20 m width). One block contained three trap–attractant combinations (trap–ethanol, trap–ethanol + MB and trap–ethanol + SA). Four replicates per trap–attractant combination were conducted, thus totaling 24 traps in each poplar plantation. All the traps inside each block were randomly distributed. Trap position was changed every two weeks in rotation. All traps were visited once a week (from May to September).

The lures were hung directly from the holes made for this purpose in the paddles, with a distance of 10 cm between each lure. Traps were tied to the poplar trunks of trees at a height of 1.5 m above the ground. Lures were attached to the trap at the highest point, and live insects were trapped in the collector at the base. Ethanol plastic bags were replaced every two weeks. Vials containing SA and MB were replaced every 40 days. The captured insects were transferred from the trap jars to 50 mL-capacity jars with the help of a brush. Live beetles were collected from the collector and taken to the laboratory. They were subsequently transported to the Plant Diseases and Pests Laboratory of the University of León, where, with the help of a magnifying glass, the insects captured in each of the treatments and plots were identified and counted.

### 2.7. Statistical Analysis

For statistical analysis of EAG data, BM SPSS Statistics 27 software (IBM Corp., Armonk, NY, USA) was used. A one-way analysis of variance (ANOVA) was conducted to evaluate differences in EAG responses among the different volatiles. When the ANOVA indicated significant differences, the Tuckey post hoc method was applied to identify which means showed significant differences. All tests were performed at a significance level of 5% (*p* ≤ 0.05), and results are presented as mean ± standard error (SE).

All statistical analyses of field test data were conducted using SPSS software, version 21. The evaluated parameters showed normal distributions and homoscedasticity, and were subjected to a one-way ANOVA, followed by Tuckey post hoc (*p* ≤ 0.05).

## 3. Results

### 3.1. Identification of Volatiles

Volatile components from the bark and leaves of USA 184-411 and I-214 *P. nigra* clones were collected by SPME and analyzed by GC-MS. The identified components and their percentages are given in Table 1, where the components are listed in order of their elution from DB-5M column. The identified VOCs composition was qualitatively measured among clone (USA 184/411 and I-214) types. In addition to qualitative differences, the headspace of the most susceptible clone showed (Table 1) an altered proportion of compounds compared to the least susceptible (I-214). An immediate observation was the presence of a high level of benzenoid derivatives, although with remarkable differences in the main components. Benzenoid derivatives were the major VOCs identified in leaves and bark for both I-214 (20.1% and 11.5%, respectively) and USA 184-411 (81% and 21.5%, respectively) clones. Salicyaldehyde accounted for 62.7% of the leaf volatiles in the USA 184-411 clone but only 3.05% of the leaf volatiles in the I-214 clone.

In the USA 184-411 bark volatiles (Table 1), the important components were salicylaldehyde (11.4%), α-copaene (5.7%), salicylic alcohol (4.2%), δ-cadinene (3.03%), and the sesquiterpene alcohols δ-eudesmol (2.70%), methyl salicylate (2.9%), β-eudesmol (2.7%), and α-bisabolol (1.3%).

### 3.2. Field Experiments

#### Captures Based on Clones and Attractants

In the Villasabariego plot, the insect captures obtained in clones I-214 and USA 184-411 with the attractant ethanol + SA are significantly greater than those collected with the other two attractants within each clone. Furthermore, within the same plot, when comparing the catches between the two clones and within the same attractant, it is observed that there are significant differences in the clone (Table 2).

In the Villoria de Órbigo plot, the catches obtained within the USA 184-411 clone with the attractant ethanol + SA were significantly higher than those obtained with the other two attractants. Furthermore, when comparing the captures obtained between clones, the ethanol + MB and ethanol + SA treatments were significantly higher than those obtained in the USA 184-411 clone with the same attractants (Table 2).

### 3.3. Electrophysiological Response of T. binodulus to Identified Compounds

Nine volatile compounds were selected for the EAG antennal detection experiment. Figure 1 shows the results of the DMS test comparison of the EAG response to the different odor stimuli. These nine compounds (salicylaldehyde, catechol, methyl benzoate, nonanal, salicylic alcohol, methyl salicylate, *trans*-β-caryophyllene, α-copaene and δ-cadinene) were selected based on leaf and bark volatiles present in both *P. nigra* clones. Adult beetle antennae elicited the strongest electrophysiological response (in relation to hexane), to salicylaldehyde (F = 6.332; df = 9.199; *p* ≤ 0.001), followed by methyl benzoate, which was not significantly different from the other compounds tested. Moreover, the insects’ antennae had a weak response to methyl salicylate, significantly lower than that shown towards salicylaldehyde and methyl benzoate (Figure 1).

## 4. Discussion

In our study, we tentatively identified 53 VOCs emitted by poplar leaves and bark. However, the total amount of salicylaldehyde released by the USA 184/411 clone in both leaf and bark volatiles is higher than that of the I-214 clone. Few species-specific semiochemicals for bark beetles affecting hardwood trees have been identified, but many studies have demonstrated that insects detect only a small portion of these compounds [3]. Salicylaldehyde is more strongly associated with *Populus* species [33]. Despite the many volatile constituents emitted by plants, many detection and monitoring efforts rely almost solely on the use of ethanol, a volatile produced by stressed trees, as an attractant [34,35]. The use of other conifer-associated volatile compounds has been met with varying success [10,35,36].

In our study, adult *Trypophloeus binodulus* of both sexes were attracted to traps loaded with salicylaldehyde + ethanol. Ethanol is a natural by-product of fermenting or dead wood and stressed trees, making it attractive to many species of bark beetles [37,38,39,40]. Low-molecular-weight alcohols, especially ethanol, are commonly associated with stress or wounds in many genera of forest trees [8,34].

The combination of the host volatiles (kairomones) ethanol and host plant compounds is attractive to many species of woodboring beetles [39,41,42,43,44]. The detection, monitoring, and mass trapping of woodboring pests in systems often exploit their chemically mediated behavior as part of an integrated pest management (IPM) strategy [45].

Species of the genus *Populus* contain salicinoids as defense compounds against insect herbivores [46,47], and salicortin is the most abundant component, which can be deglucosylated and metabolized to catechol, benzoates, and methyl salicylate derivatives [48,49].

Identifying novel semiochemicals is increasingly important as forest and plantation managers search for sustainable methods to monitor and manage insect pests, especially as a changing climate increases biotic and abiotic stress [50,51]. Utilizing semiochemicals within IPM programs can reduce the development of pesticide resistance and aid in the long-term management of native and exotic pest species. Our results support the use of salicylaldehyde to exploit the chemically mediated colonization behavior of *Trypophloeus binodulus.*

In previous studies, salicylaldehyde’s effectiveness as an attractant in insects has been demonstrated. For the *Trypodendron* genus beetles, specifically the species *T. retusum*, colonizers of aspen, greater captures were observed in those traps baited with the compound, supporting the role of salicylaldehyde as an attractive volatile compound for this species [52].

In the two plots described in this work, captures were made using the traps described above in two blocks of poplar, I-214 and USA 184/411 clones, for 19 weeks. With the attractant salicylaldehyde, greater captures were achieved in two of the analysis plots. It should be noted that, according to The Pherobase (Database of Pheromones and Semiochemicals) [53], salicylaldehyde participates in chemical communication within the order Coleoptera, which reaffirms the results obtained. These results contrast with those obtained in other tests carried out for the species *T. klimeschi* [22]. This species, discovered in China, displays specificity to *P. alba* var. *pyramidalis*, and during a field evaluation with various volatile compounds, it was observed that methyl benzoate had greater captures and, therefore, a higher attraction capacity for this species. The present work also differs from the previously described work [22] in the type of traps used; in this case, multi-funnel traps were utilized, with six plastic funnels aligned vertically.

The flights of adult Scolytidae tend to fluctuate enormously during their seasonal cycle [54]; our results indicate a minority presence during May and June, although there are some exceptions in Villasabariego, increasing notably in May, June, July, and August, the number of captures decreased again in September. For the three work plots, it should be noted that no notable flight peaks have been observed; that is, the emergence of the insect is staggered in this case. In this study, an evaluation is made of the behavior of insects of the genus *T. binodulus* in response to volatile compounds, thus verifying their possible effectiveness as a monitoring and control method. After conducting out laboratory bioassays for the identification of volatile compounds present in the bark and leaf extracts, as well as the compounds emitted by insects from another project, the possible attractants, salicylaldehyde and methyl benzoate, were evaluated in field bioassays.

It would be interesting to propose new research that focuses on providing more information following this first investigation on the control of *Trypophloeus binodulus* using kairomonal compounds in *Populus* spp. It is necessary to propose this new line of research, thus allowing for assurance regarding whether salicylaldehyde has attractant behavior or a synergistic effect with ethanol. If traps with exclusive salicylaldehyde as an attractant were effective, eliminating ethanol would allow for less maintenance of these, removing the need to change the ethanol every 15 days, as was carried out in this case. Furthermore, in this study, the captured individuals have not been sexed, which is important for future work to propose different possibilities for controlling the insect. It is also vital to know if other types of traps are more effective in capturing specimens and if the concentrations used in the vials are the most appropriate.

## 5. Conclusions

To the best of our knowledge, this study is the first to investigate both electrophysiological and behavioral responses of *Trypohloeus* spp. to their host-plant volatile compounds. Salicylaldehyde is the most EAG-active compound and is present in high quantities in the most susceptible poplar clone. Traps baited with saliciyaldehyde + ethanol significantly captured a higher number of beetles. These experiments may lead to the optimization of a synthetic lure that may be used to detect and/or monitor *Trypophloeus binodulus* populations.

## Figures and Tables

**Figure 1 insects-15-00964-f001:**
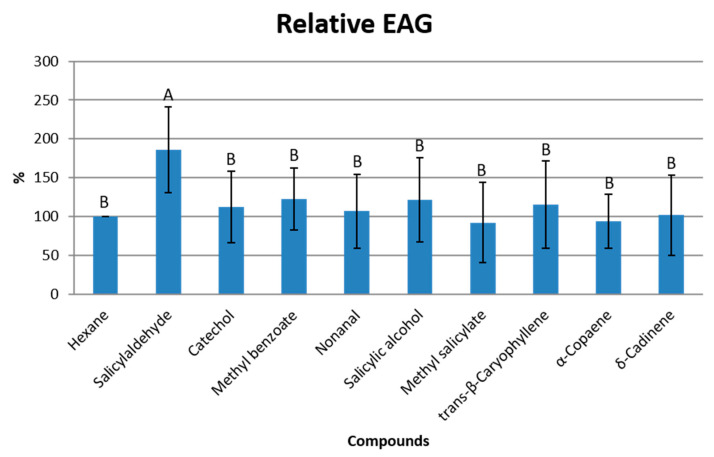
Electrophysiological responses (% ± standard error) of *T. binodulus* adults (*n* = 20) to compounds identified in Poplar (*P. nigra*) volatiles. Different letters within each group denote significant differences in the EAG response among quantities (one-way ANOVA followed by Tuckey’s post hoc test (*p* ≤ 0.05).

**Table 1 insects-15-00964-t001:** Retention times (RT), retention indices (RI), and relative abundance (Mean±SD%) of the volatile chemical composition of *Populus nigra* bark and young leaves.

	Relative Abundance (%) ^c,d^						
Compound	RT ^a^ (min)	RI-5 ^b^	Clone USA 184-411	Clone I-214	Clone USA 184-411	Clone I-214	Identification ^e^
Isobutanol	3.66	635	0.17 ± 0.07	0.28 ± 0.10	n.d.	n.d.	MS; STD
(E)-3-Hexen-1-ol	6.63	853	n.d.	n.d.	0.09 ± 0.03	0.06 ± 0.03	MS; STD
(E)-2-Hexenal	6.71	857	n.d.	n.d.	1.71 ± 0.24	0.29 ± 0.09	MS; STD
(Z)-3-Hexen-1-ol	6.89	861	n.d.	n.d.	2.04 ± 1.35	1.38 ± 0.58	MS; STD
Benzaldehyde	6.92	966	0.83 ± 0.24	0.37 ± 0.21	0.30 ± 0.20	n.d.	MS; STD
Limonene	6.97	1030	0.13 ± 0.05	0.28 ± 0.11	0.39 ± 0.09	n.d.	MS; STD
1.8-Cineole	8.19	1038	n.d	0.09 ± 0.02	0.07 ± 0.02	n.d.	MS; STD
Salicylaldehyde	6.46	1045	11.38 ± 2.77	7.90 ± 0.34	62.73 ± 14.5	3.05 ± 0.54	MS; STD
β-Ocimene	4.12	1048	0.16 ± 0.07	n.d.	0.87 ± 0.19	n.d.	MS; STD
p-Cresol	4.73	1078	2.39 ± 0.90	0.47 ± 0.17	3.18 ± 1.80	n.d.	MS; STD
Methyl benzoate	5.32	1090	0.16 ± 0.10	0.64 ± 0.25	0.48 ± 0.17	0.57 ± 0.13	MS; STD
Linalool	5.56	1099	0.07 ± 0.03	n.d.	0.15 ± 0.06	0.22 ± 0.04	MS; STD
Nonanal	8.87	1108	1.19 ± 0.82	0.13 ± 0.04	0.64 ± 0.09	0.14 ± 0.05	MS; STD
Catechol	9.11	1199	0.28 ± 0.07	0.51 ± 0.12	0.16 ± 0.07	0.12 ± 0.06	MS; STD
Salicylic alcohol	8.82	1230	4.18 ± 2.44	1.50 ± 0.33	0.29 ± 0.13	0.26 ± 0.10	MS; STD
Methyl salicylate	11.24	1294	2.89 ± 0.80	0.16 ± 0.12	0.30 ± 0.06	0.19 ± 0.08	MS; STD
Eugenol	11.35	1357	n.d.	n.d.	3.67 ± 1.17	17.26 ± 3.5	MS; STD
trans-β-Ionone	11.72	1366	n.d.	n.d.	0.23 ± 0.10	0.09 ± 0.04	MS; STD
α-Copaene	11.89	1379	5.66 ± 3.32	2.11 ± 0.54	1.07 ± 0.35	0.16 ± 0.05	MS; STD
β-Cubebene	12.00	1390	0.11 ± 0.01	n.d.	n.d.	n.d.	MS
α-Caryophyllene	12.21	1408	0.92 ± 0.17	0.13 ± 0.04	0.35 ± 0.22	n.d.	MS; STD
E-β-Caryophyllene	12.84	1417	1.82 ± 0.29	0.18 ± 0.07	0.66 ± 0.28	1.30 ± 0.25	MS; STD
α-Guaiene	13.05	1441	0.38 ± 0.20	n.d.	n.d.	0.67 ± 0.15	MS
trans-β-Farnesene	13.18	1458	0.46 ± 0.18	n.d.	0.79 ± 0.33	n.d.	MS; STD
Alloaromadendrene	13.26	1453	1.49 ± 0.58	n.d.	n.d.	1.04 ± 0.43	MS
β-Guaiene	13.32	1491	0.67 ± 0.18	n.d.	0.38 ± 0.08	n.d.	MS; STD
Valencene	14.54	1494	0.57 ± 0.17	0.57 ± 0.23	n.d.	0.14 ± 0.05	MS
E-β-Bergamotene	13.67	1580	0.04 ± 0.02	n.d.	0.49 ± 0.24	n.d.	MS
γ-Muurolene	13.79	1479	0.28 ± 0.03	n.d.	0.37 ± 0.19	0.39 ±	MS
α-Muurolene	14.88	1501	0.72 ± 0.27	n.d.	n.d.	n.d.	MS
α-Bisabolene	15.00	1503	n.d.	n.d.	n.d.	0.58 ± 0.22	MS
(E.E)-α-Farnesene	15.18	1507	0.11 ± 0.05	n.d.	0.46 ± 0.09	0.22 ± 0.05	MS; STD
β-Bisabolene	15.29	1512	0.26 ± 0.11	n.d.	n.d.	n.d.	MS; STD
Curcumene	15.41	1516	0.19 ± 0.03	n.d.	n.d.	n.d.	MS
δ-Cadinene	16.03	1527	3.03 ± 0.05	2.04 ± 0.89	0.29 ± 0.13	2.81 ± 0.55	MS; STD
α-Calacorene	17.20	1542	0.13 ± 0.02	n.d.	0.19 ± 0.09	0.27 ± 0.12	MS
trans-Nerolidol	17.65	1564	n.d.	0.05 ± 0.01	n.d.	n.d.	MS
α-Caryophyllene	18.37	1583	0.32 ± 0.20	n.d.	n.d.	0.62 ± 0.16	MS
Guaiol	18.52	1608	0.70 ± 0.32	n.d.	0.19 ± 0.13	0.24 ± 0.13	MS; STD
γ-Eudesmol	18.66	1621	0.46 ± 0.10	n.d.	n.d	n.d	MS
δ-Eudesmol	18.71	1631	2.69 ± 0.94	n.d.	2.56 ± 1.68	0.29 ± 0.22	MS
Cubenol	19.00	1639	n.d.	n.d.	n.d.	3.81 ± 1.61	MS
β-Eudesmol	19.15	1653	2.69 ± 0.41	n.d	n.d	1.80 ± 0.76	MS
α-Cadinol	19.48	1652	0.71 ± 0.28	n.d.	0.27 ± 0.09	0.86 ± 0.36	MS
α-Muurolol	20.11	1653	0.33 ± 0.05	n.d.	n.d.	n.d.	MS
Tridecanoic acid	22.13	1670	2.09 ± 0.96	n.d.	n.d.	n.d.	MS
α-Bisabolol	22.40	1675	1.33 ± 0.35	n.d.	n.d.	n.d.	MS
Methyl linoleate	30.37	1716	n.d	0.77 ± 0.09	n.d.	2.25 ± 0.57	MS
Hexadecanoic acid	34.50	1968	2.35 ± 1.00	7.46 ± 2.73	1.12 ± 0.30	6.77 ± 0.35	MS; STD
Hexatriacontane	39.88	2095	n.d	26.09 ± 7.03	n.d.	n.d.	MS
Linoleic acid	40.03	2127	2.89 ± 1.45	0.30 ± 0.10	n.d	0.40 ± 0.20	MS; STD
Oleic acid	40.62	2136	0.35 ± 0.27	0.31 ±0.14	n.d.	0.19 ± 0.02	MS; STD
Octadecanoic acid	42.12	2175	2.15 ± 0.38	1.26 ±0.39	n.d.	0.73 ± 0.10	MS; STD

^a^ Retention time on a DB-5 capillary column. ^b^ Retention indices on a DB-5 capillary column. ^c^ The values are mean ± standard deviation. n.d.: not detected; refers to a percentage below 0.05%. ^d^ Relative percentage: individual component in relation to total volatile constituents. ^e^ Identification method: MS—identification was made by comparing the mass spectrum with databases; STD—identification was made by comparisons with the commercial standard.

**Table 2 insects-15-00964-t002:** Captures in traps (mean ± standard error) from May to September 2022.

Poplar Plantations	Attractants		*T. binodulus* Captures					
			Clone I-214		Clone USA 184-411	F	df	*p*
Villasabariego	Ethanol		4.32 ± 0.80 b ^1^		7.16 ± 1.20 b	3.887	(1.36)	0.050
	Ethanol + MB		2.47 ± 0.64 b		4.21 ± 1.00 b	2.106	(1.36)	0.155
	Ethanol + SA		10.37 ± 2.03 a		25.37 ± 5.20 a	7.196	(1.36)	0.011
		F	9.848	F	13.320			
		df	(2.54)	df	(2.54)			
		*p*	<0.001	*p*	<0.001			
Villoria de Órbigo	Ethanol		5.00 ± 1.34 a		6.89 ± 1.87 b	0.673	(1.36)	0.417
	Ethanol + MB		2.16 ± 0.36 a		9.68 ± 2.42 b	9.008	(1.36)	0.005
	Ethanol + SA		2.37 ± 0.62 a		34.16 ± 7.02 a	20.320	(1.36)	<0.001
		F	2.894	F	11.488			
		df	(2.54)	df	(2.54)			
		*p*	0.064	*p*	<0.001			

^1^ Different lowercase letters indicate significant differences between attractants within the same clone and plot. Tuckey’s post hoc test (*p* ≤ 0.05). A generalized lineal model, ANOVA (one way), was used. Evaluated means parameters were normally distributed, presented homoscedasticity, and were subjected to statistical analysis (ANOVA).

## Data Availability

All the data generated from the current research are available upon reasonable request to the corresponding author.

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
