# Peer review of "Salicylic Aldehyde and Its Potential Use in Semiochemical-Based Pest Control Strategies Against Trypophloeus binodulus"

_insects, 2024, doi:10.3390/insects15120964_

Round 1

Reviewer 1 Report

Comments and Suggestions for Authors

This manuscript contains interesting results with potential practical applications for management of an important pest of poplars. The authors present fairly clear evidence that adding salicylaldehyde to ethanol-baited traps significantly increases the mean catch of Trypophloeus spp. beetles. They also show that salicyaldehyde is present in much greater concentrations in the leaves of a P. nigra clone that is attacked more heavily by Trypophloeus spp., that salicyaldehyde elicits a very strong antennal response in the beetles, and that trap catches are greater in areas of the poplar plantations where the most susceptible clone is present.  However, there are several problems with the ways the manuscript has been written that must be addressed before it it is acceptable for publication. The methods have not been described clearly enough. The method of statistical analysis is not described in sufficient detail. Much of the Discussion is redundant, repeating points already made in the Introduction.  I have asked for clarification of several points in an annotated version of the pdf and have made some editorial suggestions, though I stopped making these in the Discussion.

Comments on the Quality of English Language

Please see comments in the annotated pdf.

Author Response

This manuscript contains interesting results …….However, there are several problems with the ways the manuscript has been written that must be addressed before it is acceptable for publication. The methods have not been described clearly enough. The method of statistical analysis is not described in sufficient detail. Much of the Discussion is redundant, repeating points already made in the Introduction.  I have asked for clarification of several points in an annotated version of the pdf and have made some editorial suggestions, though I stopped making these in the Discussion

We thank the reviewer 1 for his/her kind comment concerning the significance of the reported results and their relevance to the scope of Insects. We thank the reviewer for the careful reading and the useful comments that we capitalized to improve our manuscript.

Our point to point response to yours interesting comments is shown below.

The SPME fiber was inserted into the container 4 hours after the plant material had been added to the container, but for how long did you leave the SPME fiber in the container?”

Reply: We thank the reviewer for pointing out the lack of collection time. This detail have been added in the section Material and Methods. L-159. Done

This is confusing.  The antennae were excised but then you say you used a head/antennal preparation?  Was an antennae cit off and placed across a pair of electrodes or was one elctrode in the head and the other on the tip of an antenna still attached to the head but with the tip cut off?”

Reply: We agree with the reviewer, both details have been added in the section Materials and Methods. L-184 to L-186. Done

“you said the test stimuli were puffed for 2 seconds above, now you saw 1 second?”

1 second. Done

“This description would fit better up at lines 116-120 when you first mention how you collected the beetles for the experiments.”

Our phrasing was not very clear and it has now been improved to show that we meant. Done

“you have not yet defined what SA and MB are. Please move the description iof lures and dispensers up to precede this section”

Response: We fully agree with the Reviewer. This paragraph has been rewritten in the new version of the manuscript. Done

is the bolded text on purpose or an accident?

Reply: Sorry for the mistake. Done

“Now the statement about "clone-dependent" makes sense but this paragraph needs to be reordered to clarify things to the reader, i.e., first state that the insect preferentially attacks one particular clone of P. alba pyramidalis, then discuss host volatiles”

Reply: Thanks for your suggestion! This paragraph has been moved to Field trapping section in the new version of the manuscript. Done

“Move this section up ahead of the Field trapping section.  Also, define "SA" and "MB" by adding them in parentheses after "salicyaldehyde" and "methyl benzoate" respectively.”

Reply: Thanks for your suggestion! This paragraph has been moved to the Field trapping section in the new version of the manuscript. Done

“I am not familiar with this test. Do you mean Fishers least significant difference test? Moreover, you have provided no description for the statistical analyses of the data. What was the model? Was it a generalized linear mixed model with lure and clones fixed and replicates random? Did you use SAS or R? Please provide the details.”

Reply: We agree with the reviewer, both details have been added in the section 2.7 in the new manuscript version. Done

“This entire paragraph is just repetition of material in the Introduction and can be deleted so that the Discussion starts with the main findings”

Reply: Thanks for your suggestion! This paragraph has been moved to Introductrion section in the new version of the manuscript. Done

This is the main finding of the study and should be up front in the Discussion. Most of the preceding text in the Discussion has already been stated in the introduction and can be severely trimmed or cut enitrely.

Response: We fully agree with the Reviewer. This paragraph has been rewritten in the new version of the manuscript. Done

Reply: We agree with the Reviewer about the selected compounds used for EAG bioassay. These nine compounds (salicylaldehyde, catechol, methyl benzoate, nonanal, salicylic alcohol, methyl salicylate, trans-β-caryophyllene, α-copaene and δ-cadinene) were selected based on leaf and bark volatiles present in both P. nigra clones and eugenol is not common in both. In addition, in previous tests, it does not show electroantennographic response.

Reviewer 2 Report

Comments and Suggestions for Authors

Review Report

This study investigates the role of volatile organic compounds in the colonization behavior of the poplar bark beetle (Trypophloeus spp.), a major pest of poplar (Populus spp.) trees in northern Spain. The research focuses on identifying chemical attractants from poplar clones, specifically examining the clone USA 184-411, which is highly susceptible to bark beetle infestations. The findings contribute to understanding how Trypophloeus is attracted to certain poplar clones and provide insights into potential semiochemical-based pest management strategies.

The study takes an important and novel approach by exploring the chemical mediation of bark beetle attraction to susceptible poplar clones. The identification of salicylaldehyde as a potent attractant is a valuable discovery that could contribute significantly to developing new pest management strategies.

The volatile organic compound (VOC) analysis using SPME (solid-phase microextraction) was well executed, identifying 53 compounds across various chemical classes. The breakdown of monoterpenes, sesquiterpenes, benzenoids, alcohols, aldehydes, and alkanes adds a detailed layer of understanding regarding the chemical ecology of Trypophloeus spp.

The use of electroantennography (EAG) to measure beetle responses to specific volatiles provides strong physiological evidence that salicylaldehyde plays a significant role in beetle attraction. The field trials, which confirmed the bioactivity of salicylaldehyde in attracting adult beetles, add practical relevance to the findings and support the potential for semiochemical-based pest control.

The study’s practical implications are substantial. Demonstrating the efficacy of traps baited with salicylaldehyde+ethanol in capturing more adult beetles suggests that these findings could be translated into actionable pest control measures. This offers a potential alternative to chemical insecticides, aligning with more sustainable pest management approaches.

Overall, this research represents a solid contribution to the field, and I recommend minor revisions to address the weaknesses outlined in »comments for authors«.

Comments for authors

The Materials and Methods section of the manuscript provides a detailed account of the experimental procedures used to study the response of Trypophloeus spp. to various volatile organic compounds (VOCs) and trapping techniques. However, several shortcomings could potentially limit the clarity, reproducibility, and reliability of the results. Below are some of the identified issues:

1.       The environmental conditions (temperature and light cycle) for maintaining the Trypophloeus spp. are provided, but there is no mention of the humidity levels. Relative humidity could influence insect behavior and physiology, so its absence raises concerns about the consistency of insect handling.

2.       While the insects were collected using ESCOLITRAP®, there is no description of how often the traps were checked and how long the insects were stored before being used in experiments. This could affect the condition of the insects and introduce variability.

3.       VOCs were collected for four hours, but the rationale for choosing this time frame is not provided. Variability in the duration of volatile collection could influence the concentration and profile of the emitted compounds, and this could be better justified in the methods.

4.       The description of the GC-MS analysis lacks crucial details about how the data were processed, such as the software used for compound identification and quantification. Additionally, there is no mention of calibration curves or the use of internal standards to ensure accuracy and consistency in compound quantification.

5.       Limited Information on Experimental Layout: The description of trap placement (1.5 m above the ground, 0.2 ha plots) is given, but there is no mention of how environmental factors like wind direction or proximity to other trees could affect trap efficacy. Were the traps rotated during the study to avoid positional bias?

6.       VOCs can degrade over time, so there should be information about how often the lures were replaced to maintain consistent attractiveness over the course of the experiment.

7.       The methods mention that two replicates per trap–attractant combination were conducted, but it is unclear whether these replicates are sufficient for robust statistical analysis. Typically, more replicates would be expected in field experiments to account for environmental variability.

Author Response

Comments to Reviewer 2.

We thank the reviewer for his careful reading and helpful comments, which we have used to improve our manuscript. We really appreciate the reviewer’s comments and the suggestions.  

The environmental conditions (temperature and light cycle) for maintaining the Trypophloeus spp. are provided, but there is no mention of the humidity levels. Relative humidity could influence insect behavior and physiology, so its absence raises concerns about the consistency of insect handling

Reply: We thank the reviewer for pointing out the lack of that lab condition data. Done

  1. While the insects were collected using ESCOLITRAP®, there is no description of how often the traps were checked and how long the insects were stored before being used in experiments. This could affect the condition of the insects and introduce variability”.

Reply: We agree with the reviewer, both details have been added in the section Materials. Done

  1. VOCs were collected for four hours, but the rationale for choosing this time frame is not provided. Variability in the duration of volatile collection could influence the concentration and profile of the emitted compounds, and this could be better justified in the methods”.

Reply: Thank you for this crucial comment. Sampling times from 1 to 4 h were tested; mainly due to the low amounts of volatile released by poplar bark, but in the end it was found that 1 h was sufficient to obtain good results. This information has been included in the new version of the manuscript. Done

  1. The description of the GC-MS analysis lacks crucial details about how the data were processed, such as the software used for compound identification and quantification. Additionally, there is no mention of calibration curves or the use of internal standards to ensure accuracy and consistency in compound quantification”.

Reply: We fully agree with the reviewer, but normally the percentage of each peak in the GC chromatogram is a reflection of the proportion of a specific analyte present, so the area peak will be based on the number of counts taken by the mass spectrometer quadrupole at the point of retention. Really our data showed a qualitative analysis which aids in the identification of VOC components, whereas quantitative analysis allows for the precise determination of their quantities. The objective was not the exact determination of the quantity of each of them, which in this case does require the inclusion of an internal standard or calibration curves, which would correspond to a quantitative analysis. In most of the similar studies, (I attach some references) the proportion in % of each component, relative area Peak or relative abundance of the total of the peaks detected by the mass spectrum is presented

https://doi.org/10.3390/insects15100739

https://doi.org/10.3390/insects13090840

https://doi.org/10.3390/insects15060454

https://doi.org/10.3390/insects15060402

  1. Limited Information on Experimental Layout: The description of trap placement (1.5 m above the ground, 0.2 ha plots) is given, but there is no mention of how environmental factors like wind direction or proximity to other trees could affect trap efficacy. Were the traps rotated during the study to avoid positional bias?

Reply: The reviewer is correct in pointing out that poplar plantation characteristics are very important in field trials. Every two weeks a review of the captures was made and the traps were rotated to eliminate the effect of the trap position in the plot. That information has been included in the revised manuscript. Done

6.”VOCs can degrade over time, so there should be information about how often the lures were replaced to maintain consistent attractiveness over the course of the experiment”.

Reply: We agree with the reviewer, maybe it was not very clear in the original text that the lures were changed every two weeks (ethanol) and 40 days (SA and MB). This information has been included in the revised manuscript. Done

7.”The methods mention that two replicates per trap–attractant combination were conducted, but it is unclear whether these replicates are sufficient for robust statistical analysis. Typically, more replicates would be expected in field experiments to account for environmental variability”

Reply: Unfortunately, we could not record the captures of the third trial because, during summer time when we arrived at the experimental site, there had been a thinning of the poplar trees, and we had to abandon it. The traps were divided into two blocks (one in the I-214 clone and one in the USA 184-411 clone) of an area of 0.4 ha (40 m length × 20 m width). Four replicates per trap–attractant combination were conducted, thus totaling 24 traps in each poplar plantation. All the traps inside each block were randomly distributed. Trap position was changed every two weeks in rotation. Despite this, 48 traps were installed in two different poplar plots ( 57 km from each other) during 5 months with a weekly revision of the captures, we believe that this is sufficient data for statistical analysis.

Reviewer 3 Report

Comments and Suggestions for Authors

General comments

Abstract. The abstract should be a total of about 200 words maximum. Currently, the abstract contains 288 words. Please, shorten the abstract to follow the author's guide of the journal.

Table 1 was not referred in the text.

Specific comments

Line 30. Please, change (Salicaceae) to (Malpighiales: Salicaceae)

Line 38. Please, insert “to” between the words “than any”.

Line 42. Space is needed “abark”.

Line 59.”During the recognition selection and colonization process…” In my understanding, two commas are missing.

Lines 77-78. “This phenomenon has profound ecological implications for the insect, its interaction with the environment, and its life cycle.” Please, provide a little more information on the implications of this phenomenon.

Line 89. “…(northwest Spain).In the poplar groves…” A space between sentences is missing.

Line 90. The space is missing between the words “greatvariety

Line 91. “This observation is problematic since….” I would use rise concern or another synonym instead of “is problematic”

Lines 139-140. Despite the fact that these two types of SPME fiber coatings are widely used, please provide the full name of the polymers, i.e. PDMS and DVB.

Lines 141-145. I have some questions in respect to the sampling procedure. The plant material was placed into a glass container. Was this glass container closed, forming headspace, or left open?

“…then we inserted an SPME needle into the beaker and positioned the fiber away.” It is not clear to me what you mean by that. Usually after a sampling, SPME fiber is moved back into a needle.

Samples were stored in vials at −20 °C until…” Which samples do you mean? VOCs collected on SPME fibers?

Line 147. “Loaded SPME fibers were analyzed…” I would simply write Analyses were carried out by Thermo FOCUS GC …

Lines 152-153. Description of the GC protocol has mistakes “The injection port was maintained at 280 °C for 1 min, increased by 5 °C min−1 to 250 °C and held for 15 min.” How could you increase the temperature from 280 ° C to 250 °C? Probably you were describing oven temperature program. Please correct the description.

Lines 202-204. It looks like control traps without semiochemicals were not used. Why?

It is not clear how the lures containing Ethanol+MB and Ethanol+SA were formed? Were the Ethanol dispenser and MB dispenser placed near each other in the trap?

Table 1. Please, make the title of the table more informative. In addition to RT and RI, the relative abundance of VOCs is also presented.

Table 1, line Isobutanol. Based on the retention time of this compounds, RI value is miscalculated. It can’t be 1092.

Please change “(Z)3-Hexen-1-ol” to (Z)-3-Hexen-1-ol

p-Cresol character p has to be in italic

What do numbers in red mean?

Lines 282-283. “…where five of them are common to the Salicaceae taxonomic group.” What do you mean by “common”? How do you define “common”?

Lines 318-319. “ Our field test demonstrated that Trypophloeus spp are not attracted to salicylaldehyde alone…” Unfortunately, I can’t find data in the manuscript supporting this statement.

Line 332. “…being effective in the function of the results obtained [55].” Please explain what is meant by “in the function of the results”.

Line 334. Please change “For the Trypodendron spp. genus, …” to For the beetles of Trypodendron genus,… Salicylaldehyde or other semiochemicals do not affect genus, which is a taxonomic unit. Semiochemicals affect organisms, in your case, beetles from the genus Trypodendron.

Lines 365-367. “On the one hand, since there is no trap in these tests with an attractant that exclusively contains salicylaldehyde, it must be thought that this has a synergistic effect with ethanol.” You can’t conclude that it must be a synergistic effect without having catches from the trap baited with exclusively salicylaldehyde. You may propose tests to clarify the mode of action, therefore, please rewrite the sentence.

Line 390. Did the authors have funds for this work?

Lines 380-381. The last sentence in the conclusions looks like incomplete.

Author Response

Comments to Reviewer 3.

We thank the reviewer for his/her careful reading and helpful comments, which we have used to improve our manuscript. Please find below a point-by-point list of the answers to your comments.

General comments

“Abstract. The abstract should be a total of about 200 words maximum. Currently, the abstract contains 288 words. Please, shorten the abstract to follow the author's guide of the journal”.

Reply: Dear reviewer, thanks for your comment, we have corrected the abstract and the new version contains 201 words. Done

Table 1 was not referred in the text.

Reply: Table 1 is referred in the new manuscript version. Line 234

Specific comments

Line 30. Please, change (Salicaceae) to (Malpighiales: Salicaceae)”

Reply: The order is included in the new manuscript. Done

Line 38. Please, insert “to” between the words “than any”.

Reply: Thanks. to is included in the new manuscript. Done

Line 42. Space is needed “abark”.

Reply. Thank you very much for your correction! It does not appear in the new version because we had to delete the complete sentence, as the original summary was over 200 words long.

Line 59.”During the recognition selection and colonization process…” In my understanding, two commas are missing.

Reply. It does not appear in the new version because we had to delete the complete sentence, as the original summary was over 200 words long. Done

Lines 77-78. “This phenomenon has profound ecological implications for the insect, its interaction with the environment, and its life cycle.” Please, provide a little more information on the implications of this phenomenon.

Reply. Although the loss of value of the wood is very important due to the presence of these stains, the reviewer is right about the importance of this sentence. Unfortunately, there is not enough literature about Trypophloeus bioecology, phenology even the microbiota associated with this genus is still unknown, so we have decided to delete the sentence. Done

Lines 89 Line 90 and 91.

Reply. Thanks for your correction! It does not appear in the new version. Done.

Lines 139-140. “Despite the fact that these two types of SPME fiber coatings are widely used, please provide the full name of the polymers, i.e. PDMS and DVB”.

Reply. We agree with the reviewer that it was our oversight not to mention the chemical composition of SPME fibers. The PDMS and DVB complete names are included. Done

Lines 141-145. “I have some questions in respect to the sampling procedure. The plant material was placed into a glass container. Was this glass container closed, forming headspace, or left open?” “…then we inserted an SPME needle into the beaker and positioned the fiber away.” It is not clear to me what you mean by that. Usually after a sampling, SPME fiber is moved back into a needle.

Reply: Thanks for your suggestion! This paragraph has been rewritten in the new version of the manuscript. Done

” Samples were stored in vials at −20 °C until…” Which samples do you mean? VOCs collected on SPME fibers?

Reply: The Reviewer is right and the volatile collection section is now significantly improved. Done

Line 147. “Loaded SPME fibers were analyzed…” I would simply write Analyses were carried out by Thermo FOCUS GC …

Reply: We have rewritten this section accordingly. Done

Lines 152-153. Description of the GC protocol has mistakes “The injection port was maintained at 280 °C for 1 min, increased by 5 °C min−1 to 250 °C and held for 15 min.” How could you increase the temperature from 280 ° C to 250 °C? Probably you were describing oven temperature program. Please correct the description.

Reply: Sorry for the mistake, of course it was the oven program temperature. We have rewritten this section accordingly. Done

Lines 202-204. It looks like control traps without semiochemicals were not used. Why?

Reply: Previous field trials during 2021 springtime indicated that the escolitrap traps, without any bait or lure, does not catch any beetle.

It is not clear how the lures containing Ethanol+MB and Ethanol+SA were formed? Were the Ethanol dispenser and MB dispenser placed near each other in the trap?

Reply: L221-222: The lures were hung directly from the holes made for this purpose in the paddles and 10 cm from each other. Done

Table 1. Please, make the title of the table more informative. In addition to RT and RI, the relative abundance of VOCs is also presented.

Reply: Done

Table 1, line Isobutanol. Based on the retention time of this compounds, RI value is miscalculated. It can’t be 1092.

Reply: Sorry for the error, the Reviewer is right, the isobutanol is the first peak in the chromatogram and its RI is 635. Done

Please change “(Z)3-Hexen-1-ol” to (Z)-3-Hexen-1-ol

Reply: Done

p-Cresol character p has to be in italic

Reply: Done

What do numbers in red mean?

Reply: Sorry for the error. This is probably due to Microsoft's text review option. It does not appear in the new version. Done

Lines 282-283. “…where five of them are common to the Salicaceae taxonomic group.” What do you mean by “common”? How do you define “common”?

Reply: The Reviewer is right, the word “common” is not scientific at all. Our phrasing was not very clear and it has now been improved. Done

Lines 318-319. “ Our field test demonstrated that Trypophloeus spp are not attracted to salicylaldehyde alone…” Unfortunately, I can’t find data in the manuscript supporting this statement.

The reviewer is right to point out that his is unable to find data on traps loaded with salicylaldehyde alone, because these are data from 2023 and could not be repeated during the current year's trials. So, we decided not to include them in the manuscript. We have removed this sentence from the initial text and hope to be able to answer it in future research. Done

Line 332. “…being effective in the function of the results obtained [55].” Please explain what is meant by “in the function of the results”.

Thanks to the reviewer for the suggestion. This part of the conclusion has been removed in the final document. Done

Line 334. Please change “For the Trypodendron spp. genus, …” to For the beetles of Trypodendron genus,… Salicylaldehyde or other semiochemicals do not affect genus, which is a taxonomic unit. Semiochemicals affect organisms, in your case, beetles from the genus Trypodendron.

Thanks to the reviewer for the suggestion. The world beetles has been included.

Lines 365-367. “On the one hand, since there is no trap in these tests with an attractant that exclusively contains salicylaldehyde, it must be thought that this has a synergistic effect with ethanol.” You can’t conclude that it must be a synergistic effect without having catches from the trap baited with exclusively salicylaldehyde. You may propose tests to clarify the mode of action, therefore, please rewrite the sentence.

Thanks to the reviewer for the suggestion. I believe that in the text we indicate that the synergistic effect is against traps that exclusively use ethanol as attractant. Field data in all trials indicate that captures are increased by the presence of salicylaldehyde.

Line 390. Did the authors have funds for this work?

Unfortunately, funding for the study was not available during 2022 and 2023 years. For this year, 2024, we planned to continue field trials using only salicylaldehyde (SA) and optimizing the SA lures, to avoid the use of ethanol for monitoring o mass trapping purposes. But we are still waiting for the result of the research project application.

Lines 380-381. The last sentence in the conclusions looks like incomplete.

Thanks to the reviewer for the suggestion. Our phrasing was incomplete and it has now been improved.

Reviewer 4 Report

Comments and Suggestions for Authors

General evaluation and recommendation:

The goal of the current study is to identify novel attractants for Trypophloeus in the volatile bouquet of  bark and leaves of two Populus nigra clones that are known to be different in their infestation by this species. The topic of the manuscript is important and interesting from an applied perspective.

The authors set out to test the hypothesis that Trypophloeus spp. are attracted to compounds released by the host.  

To test this hypothesis the authors analyze the volatile emission of these clones, compare the antennal responses of Trypophloeus species between host-related volatiles  and test the efficiency of traps loaded with candidate attractants at two field locations.  The choice of methods are well supported, but the design and execution of experiments and analysis are rudimentary and not sufficient to draw biologically relevant conclusions. I do not recommend the manuscript for publication without further experiments, additional data analysis and revision of the text.

Major concerns:

Insects: There was no species level identification and the sex and age of  insects trapped or used in the electrophysiological experiments were not considered. The species, physiological state and sex of the insects can significantly influence the antennal responses.

Volatile analysis:

I is stated in the method description that there were two types of SPME fibers used. These fibers differ both in their coating material (PDMS and PDMS/DVB) and coating thickness.

There is no information provided on which fiber was used in case of which sample and samples collected using different adsorbents are evidently not comparable.

The method description and results do not provide information on how many biological replicates were sampled and analyzed. The relative abundance is not shown as a mean or standard deviation in  table 1 but as a single value. Based on these results one must assume that there was no effort done to repeat the volatile collections. Furthermore there was neither a blank sample of the volatile collection device nor the SPME fiber used thus contaminations cannot be excluded. 

As volatile profiles of two individuals from the different genotypes of the same species are compared we have no information if the differences between the volatile collections are due to individual differences between the biological samples or general differences between the two clones.

Even more, it cannot be excluded that difference between the volatile profiles are mostly due to the difference between the SPME fibers used. A single volatile collection cannot be assumed to reliably represent an entire group of samples.

But these differences are interpreted as biologically relevant and used for selecting compounds to test with electroantennography.

Additionally the tentative identification using GC-MS requires the analysis of results gathered on at least to GC-columns that have different polarity.

The description of volatile constituents in the results is not in line with the table showing the results and retention index of synthetic standards and references from the retention index databases are needed to evaluate the identification.

Data analysis:

The statistical methods are not described in the materials and methods. There is no information provided if all assumptions of ANOVA were tested or the rationale behind the selection of methods. Information should be provided also about the programs used. There is no statistics done on the volatile analysis results but results are interpreted as significant.

Electroantennography:

The species and sex and physiological state of individuals is not taken into account.  The base of selecting stimuli is not clearly supported based on the chemical analysis.

The amplitude of an electroantennographic response is not necessarily comparable between compounds. The vapor pressure of compounds must be taken into account when EAG is used. Without a description of statistical methods and tests of ANOVA assumptions the statistical significance of the findings are hard to interpret.

Field experiments: 

The experiments were done at two locations and in two blocks at each location where there were two replicates per the trap-type per block. Without the adequate description of the statistical methods and test used to check assumptions for ANOVA the results are very hard to interpret. It is not clear if the same lures were used throughout the season. Dispensers must regularly be changed on the field to make sure that they are emitting the compounds. This is especially a concern for compounds that are more volatile such as ethanol. The same trap-type is in the same position throughout the season which means the differences between trap caches can be both due to the attractant used and the position of the trap.

Discussion:

Several statements of the discussion are not well supported by the data acquired and the references provided. It should be substantially rewritten based on additional experiments and data analysis.

Summary: The chemical analysis in its current form is insufficient. Further experiments are needed to verify the identification of poplar volatile emissions and to reliably compare the composition of the volatile profile of the two clones. The field experiment should also be repeated in an experimental design  where trap positions are more randomized and replicate numbers are higher. The analysis of the data should be redone to verify assumptions of the statistical tests are met  and the description of statistical methods should be provided.

Minor comments:

L18 chemicals term is usually used for synthetic substances

L28-29 grammar “associated with the highest susceptible poplar clone”-->  associated with the more susceptible poplar clone.

L32 only one clone is mentioned so far

L38 It is very hard to understand what does a higher response mean. If we compare two different compounds it does not necessary mean that the one that elicits a higher antennal response will also elicit a stronger behavioral response.

L51 grammar: delete" for their use"

L52 This is true for plant species. But for varieties that are created very recently on an evolutionary timescale  it is not so neccessarily true. Is there a reference supporting this statement?

L52 physiological status

L55-56 Which senses are diminished exactly?

L59-62  the meaning of this sentence is not clear

L78 without specifying the ecological implications the statement is vague

L81-82 It is not necessary that the reason of this difference is a choice based on olfaction. It can be a choice driven by olfactory cues but it can also be driven by other sensory inputs such as gustatory or tactile cues. It is also possible that the observed difference is due to performance on the host.

L82 Is there a reference for clone-dependent choice patterns to exist in bark beetles?

L89 .In → . In

L90 great variety

L91 reference missing for statement

L95 Species specific and common compounds should be defined.

L102 Does thermal desorption refer to that of the SPME? If yes it should be rephrased otherwise the reader expects the use of TDU as well.

L118-119 Taxonomic identification should be described in more detail. If sexes were inspected why are they not shown in the results separately?

L122 CAS numbers or UPAC names needed, all compound names should be formatted using the same naming system.

L139: samples collected with  two fibers that have different coating material and coating thickness cannot not be meaningfully compared. Which volatile sample  was collected with which fiber?

L44 Which other assays were SPME samples used in? There are no EAG or behavioral assays described in the manuscript where the volatile samples were used.

L151 This temperature 280 oC is higher than the temperature used to condition the fibers thus it is possible to have carryover between samples. 

L152-153 there is a mixup between the inlet temperature and the oven temperature program. The sentence in this way is not understandable

L155 technically it is not the  entire available range

L204 two replicates are not enough to exclude the effects of positions.

L206 the positions should be randomized throughout the study.

L211 reference missing for identification

L226 There is only one relative abundance value calculated, which cannot be used both quantitatively and semiquantitatively. The GCMS cannot provide quantitative information if there is no dose-response curve calculated for each compound. SPME is not suitable for quantitative comparison. A semiquantitative comparison is not possible in this case because the  different types of adsorbents have different affinity to the volatile components.

L230-232 alpha-elemene and gamma cadinene are not shown in table 1, δ-Cadinene does not account for 7.8-8.6% but 3.03-2.04% according to the table

Table 1. has several red cells

L235 Belongs to methods. Why is it furthermore?

L238 according to the table not all compounds identified were matched with synthetics. A supplementary table is needed where the comparison with the retention index of synthetics is shown to show how retention indexes are matching.

L242, This is the only subheading in the entire manuscript, were there other field trials?

L244 Based on what test is it significant? Was the homogeneity of variance and normality tested? Was the effect of trap position tested ?

L243-247 The format of this part is different. Why is it bold?

L243-247 This sentence is not easy to understand.

L247 Table 184?

Table 2 needs to be reorganized or separated into two tables. THe meaning of F, df and P( which should be small p) needs to be described based on the test used.

L255 reference needed for the DMS test and p is small.

L265 which statistics was used?

Figure 1: Why is hexane included in the statistics? If hexane has a standard deviation of 0 and all other compounds non-zero the homogeneity of variance is surely not true. 

L282 tentatively identified

L283 all latin names should be in italics

L286 Very vague statement: what is the nature of the compounds that suggests it exactly? 

L316 low molecular weight alcohol is a very loosely defined group.  This is the first point where the reason why ethanol is used is specified, this should have been described in the introduction

L341 which is the third plot?

Author Response

Comments to Reviewer 4.

We thank the reviewer 4 for his/her major and minor concerns, suggestions and comments. Our point to point response to the comments of the Reviewer 4 is shown below.

Major concerns:

Insects: There was no species level identification and the sex and age of insects trapped or used in the electrophysiological experiments were not considered. The species, physiological state and sex of the insects can significantly influence the antennal responses.”.

Unfortunately, there is not bibliography on the Trypophloeus species present in the Iberian Peninsula and therefore we have sent adults to two world-renowned Scolytidae taxonomists for species identification. We are waiting for the results.

We agree with the reviewer's suggestion of the incidence of using field-caught insects for laboratory bioassays. However, in our case, the electroantennographic response is presented in relative and not absolute values. In this way, we avoid the effect that the physiological state of the insect has on the results.

On the other hand, the use of adult insects from infested plots for EAG bioassays is very common in other similar works, for example:

https://doi.org/10.3390/insects15030162

https://doi.org/10.3390/insects14120911

https://doi.org/10.3390/insects14070627

https://doi.org/10.3390/insects13070655

Volatile analysis:

“I is stated in the method description that there were two types of SPME fibers used. These fibers differ both in their coating material (PDMS and PDMS/DVB) and coating thickness. There is no information provided on which fiber was used in case of which sample and samples collected using different adsorbents are evidently not comparable. The method description and results do not provide information on how many biological replicates were sampled and analyzed. The relative abundance is not shown as a mean or standard deviation in table 1 but as a single value. Based on these results one must assume that there was no effort done to repeat the volatile collections. Furthermore there was neither a blank sample of the volatile collection device nor the SPME fiber used thus contaminations cannot be excluded. As volatile profiles of two individuals from the different genotypes of the same species are compared we have no information if the differences between the volatile collections are due to individual differences between the biological samples or general differences between the two clones. Even more, it cannot be excluded that difference between the volatile profiles are mostly due to the difference between the SPME fibers used. A single volatile collection cannot be assumed to reliably represent an entire group of samples. But these differences are interpreted as biologically relevant and used for selecting compounds to test with electroantennography”.

Reply. We agree with the reviewer that it was our oversight not to mention the SPME fibers finally used for volatiles collection. This “volatile collection” section has been rewritten in the new version of the manuscript. In this new version, the fiber used for collection, the number of replicates per clone or the additional working conditions, which we probably did not adequately explain in the original manuscript.

Table 1 values are not the results from a single volatiles collection and injection on GC-MS, are means from 3-fold replicates (Material and Methods section). Many samples from different poplar clones (USA 184/411 and I-214) from two different poplar plantations were used. In addition, of course, before sampling, the SPME fibers were conditioned for 30 min at 250 °C in a gas chromatograph injection port depending on the temperature and conditioning time recommended by SPME PDMS/DVB fiber suppliers. The most prevalent contaminants in the SPME blank procedure are usually siloxanes, 1,9-nonanediol, phthalates, long-chain hydrocarbons, phenols and compounds from adhesives. Contaminants are unlikely to be hydrocarbon sesquiterpenes or benzenoid derivatives.

Plants constitutively emit volatile organic compounds (VOCs) from flowers, bark, associated microorganism, leave or roots. In addition the volatile profile release by a plant is influenced by abiotic factor, stress conditions, season or the year, soil type, even daily period and presence of herbivores or pathogens. However, that was not the purpose of our study. Our aim was to investigate why under field conditions, USA 184/411 poplar clone are the most susceptible to attack by beetle species of the genus Trypophloeus.

“Additionally the tentative identification using GC-MS requires the analysis of results gathered on at least to GC-columns that have different polarity”.

Years ago, the lack of GC peak compilation software or a solid literature (Adams R. 2007) on the elution order of volatile components in column chromatography required tools such as the use of columns of different polarity. But today, for plant volatile components identification studies, databases as:

https://pubs.aip.org/aip/jpr/article-abstract/40/4/043101/951258/Retention-Indices-for-Frequently-Reported?redirectedFrom=fulltext

Or using the same GC-MS conditions used by Robert Adams in his book (that we have used for volatile analysis): https://www.researchgate.net/publication/283650275_Identification_of_Essential_Oil_Components_by_Gas_ChromatographyQuadrupole_Mass_Spectroscopy

The description of volatile constituents in the results is not in line with the table showing the results and retention index of synthetic standards and references from the retention index databases are needed to evaluate the identification”.

Reply: Thanks to the reviewer for the suggestion. We have rewritten this section accordingly. Done

Data analysis:

The statistical methods are not described in the materials and methods. There is no information provided if all assumptions of ANOVA were tested or the rationale behind the selection of methods. Information should be provided also about the programs used. There is no statistics done on the volatile analysis results but results are interpreted as significant.

Dear reviewer, it is a generalized lineal model, ANOVA (one-way), with a  Fisher's LSD test. The program used was SPSS, version 21 (IBM SPSS Statistics). The details have been added in the new manuscript version with a new 2.7 Statistical Analysis section.

Electroantennography:

The species and sex and physiological state of individuals is not taken into account”. 

Reply: Answered in answer 1

“The base of selecting stimuli is not clearly supported based on the chemical analysis.”

Reply: These nine compounds (salicylaldehyde, catechol, methyl benzoate, nonanal, salicylic alcohol, methyl salicylate, trans-β-caryophyllene, α-copaene and δ-cadinene) were selected based on leaf and bark volatiles present in both P. nigra clones

“The amplitude of an electroantennographic response is not necessarily comparable between compounds. The vapor pressure of compounds must be taken into account when EAG is used”.

Reply: The EAG response is presented in relative and not absolute values. The absolute EAG response (mV) to each stimulus was divided by the mean response to the two nearest controls (hexane) to compensate for solvent and to compensate for the decrease of the antennal responsiveness during the experiment. In this way, we avoid the effect that the physiological state of the insect has on the results. For years, practically all the papers published in which the EAG bioassay appears, none of them consider the different vapor pressures between the components as a variable to be studied, nor the support (filter paper) or filter paper area, the relative humidity of the laboratory or so many variables that could influence the vaporization of the compounds used.

Without a description of statistical methods and tests of ANOVA assumptions the statistical significance of the findings are hard to interpret.

The details have been added in the new manuscript version with a new 2.7 Statistical Analysis section.

Field experiments:

The experiments were done at two locations and in two blocks at each location where there were two replicates per the trap-type per block. Without the adequate description of the statistical methods and test used to check assumptions for ANOVA the results are very hard to interpret. It is not clear if the same lures were used throughout the season. Dispensers must regularly be changed on the field to make sure that they are emitting the compounds. This is especially a concern for compounds that are more volatile such as ethanol. The same trap-type is in the same position throughout the season which means the differences between trap caches can be both due to the attractant used and the position of the trap.

Thanks for the suggestion. Information on the statistical analysis has been provided in a 2.7 Statistical Analysis section.

Discussion:

Several statements of the discussion are not well supported by the data acquired and the references provided. It should be substantially rewritten based on additional experiments and data analysis.

Reply: Thanks to the reviewer for the suggestion. We have rewritten this section accordingly. Done

Minor comments:

L18 chemicals term is usually used for synthetic substances

Reply: Thanks to the reviewer for the suggestion. The term has been changed. Done

L28-29 grammar “associated with the highest susceptible poplar clone”-->  associated with the more susceptible poplar clone.

Thanks for the suggestion. The words has been corrected. L-27. Done

L32 only one clone is mentioned so far

Thanks for the suggestion. The two clones has been mentionated. L-32. Done

L38 It is very hard to understand what does a higher response mean. If we compare two different compounds it does not necessary mean that the one that elicits a higher antennal response will also elicit a stronger behavioral response.

Reply: Thanks to the reviewer for the suggestion. The term has been changed by “significantly higher”. Done

L51 grammar: delete" for their use"

Thanks for the suggestion. The words "for their use" has been deleted. Done

L52 This is true for plant species. But for varieties that are created very recently on an evolutionary timescale it is not so neccessarily true. Is there a reference supporting this statement?

L52 physiological status

Thanks for the suggestion. The words "status" has been deleted. Done

L55-56 Which senses are diminished exactly?;  L59-62  the meaning of this sentence is not clear.;

Reply: These sentences has been synthesized to facilitate understanding . Done

L78 without specifying the ecological implications the statement is vague

L81-82 It is not necessary that the reason of this difference is a choice based on olfaction. It can be a choice driven by olfactory cues but it can also be driven by other sensory inputs such as gustatory or tactile cues. It is also possible that the observed difference is due to performance on the host.

Reply: Thanks to the reviewer for the suggestion. In accordance with your suggestions and those of other reviewers, the introduction has been rewritten in the new version of the manuscript. 

L82 Is there a reference for clone-dependent choice patterns to exist in bark beetles?

Reply: The introduction has been rewritten in the new version of the manuscript. Done

L89 .In → . In

Thanks for the suggestion. A blank space has been added

L90 great variety

Thanks for the suggestion. A blank space has been added between “great” and “variety”

L91 reference missing for statement

Reply: Garnica J. “El Escarabajo Trypophloeus Sp (Bark Beetle)”, Un Mal Enemigo Para Los Chopos Available online: https://www.garnica.one/blog/el-escarabajo-trypophloeus-sp-bark-beetle-un-mal.html (accessed on 29 May 2024).

L95 Species specific and common compounds should be defined.

Thanks for the suggestion. The reviewer is right “common”, we believe that common is not an appropriate term and have been deleted and species specific is included in the reference

L102 Does thermal desorption refer to that of the SPME? If yes it should be rephrased otherwise the reader expects the use of TDU as well.

Reply: Exactly. Volatiles are trapped inside the fibre by an equilibrium process, resulting in a concentrated sample of the headspace. The entire fibre must be inserted into the injection port of a GC, and thermally desorbed by the heat of the injection port, resulting in a rapid transfer of all absorbed components on to the GC column. Thanks for the suggestion. We have rewritten this section accordingly.

L118-119 Taxonomic identification should be described in more detail. If sexes were inspected why are they not shown in the results separately?

Reply: Unfortunately we only know the genus (Trypophloeus spp.), as we have indicated in the previous answer, adult males and females have been sent to two recognized taxonomists of Scolytidae. Although they were sexed in the field, we decided to treat them all as adults because we do not know the exact morphology of this species.

L122 CAS numbers or UPAC names needed, all compound names should be formatted using the same naming system.

Reply. To avoid long and tedious names in commercial chemical compounds, the official IUPAC naming recommendations are not always followed in practice, except when it is necessary to give an unambiguous and absolute definition to a compound. The CAS numbers are the most used unique substance identifiers, but the compounds we have used as standards or for EAG bioassay are commercial and accessible through any chemical database.

L139: samples collected with two fibers that have different coating material and coating thickness cannot not be meaningfully compared. Which volatile sample  was collected with which fiber?

Reply: At the beginning two fibers were tested, but finally just the fiber were 1 cm long and consisted of fused silica supports and 100 μm thick PDMS (Polydimethylsiloxane) coating. We have rewritten this section accordingly.

L44 Which other assays were SPME samples used in? There are no EAG or behavioral assays described in the manuscript where the volatile samples were used.

Reply: Unfortunately our GC-EAD is not active. We decided to test the EAG active compounds directly under field conditions.

L151 This temperature 280 oC is higher than the temperature used to condition the fibers thus it is possible to have carryover between samples. L152-153 there is a mixup between the inlet temperature and the oven temperature program. The sentence in this way is not understandable

Reply: We regret both mistakes. The transfer line temperature was 280ºC, but the injector port temperature was 250ºC, and of course, the program was oven and not injector. We have rewritten this section accordingl.

L155 technically it is not the entire available range

Reply. The exact mass range was 50 to 350 m/z. Done

L204 two replicates are not enough to exclude the effects of positions. L206 the positions should be randomized throughout the study.

Thanks for the suggestion. During 2022 field test (manuscript results), trap positions were changed every two weeks in rotation. For 2023 and 2024 years (results not included in the manuscript) traps were not rotated, the few replicates are complemented by the repetition of the field test for 2 years. We have rewritten this section accordingly. Done

L211 reference missing for identification

L226 There is only one relative abundance value calculated, which cannot be used both quantitatively and semiquantitatively. The GCMS cannot provide quantitative information if there is no dose-response curve calculated for each compound. SPME is not suitable for quantitative comparison. A semiquantitative comparison is not possible in this case because the  different types of adsorbents have different affinity to the volatile components.

Reply: We fully agree with the reviewer, but normally the percentage of each peak in the GC chromatogram is a reflection of the proportion of a specific analyte present, so the area peak will be based on the number of counts taken by the mass spectrometer quadrupole at the point of retention. Really our data showed a qualitative analysis which aids in the identification of VOC components, whereas quantitative analysis allows for the precise determination of their quantities. The objective was not the exact determination of the quantity of each of them, which in this case does require the inclusion of an internal standard or calibration curves, which would correspond to a quantitative analysis. In most of the similar studies, (I attach some references) the proportion in % of each component, relative area Peak or relative abundance of the total of the peaks detected by the mass spectrum is presented

https://doi.org/10.3390/insects15100739

https://doi.org/10.3390/insects13090840

https://doi.org/10.3390/insects15060454

https://doi.org/10.3390/insects15060402

L230-232 alpha-elemene and gamma cadinene are not shown in table 1, δ-Cadinene does not account for 7.8-8.6% but 3.03-2.04% according to the table

Reply: Our phrasing was not very clear and it has now been improved to show the data from Table 1.

Table 1. has several red cells

Thanks for the suggestion. It's a mistake. The numbers in Table 1 has been corrected in black colour

L235 Belongs to methods. Why is it furthermore?

L238 according to the table not all compounds identified were matched with synthetics. A supplementary table is needed where the comparison with the retention index of synthetics is shown to show how retention indexes are matching.

Reply: Unfortunately, not all compounds are commercially available or the price of some sesquiterpenes is prohibitive. In many cases, commercial sesquiterpenes have purities below 80%, making them unfeasible for chromatographic analysis. The used column GC program is the one used by Adams R.P. in his book and review, to confirm the exact elution order in the same column (5% phenylsiloxane).

L242, This is the only subheading in the entire manuscript, were there other field trials?

Thanks for the suggestion. The subheading has been integrated into the text

L244 Based on what test is it significant? Was the homogeneity of variance and normality tested? Was the effect of trap position tested ?

Significance was based on Fisher's LSD test. Means parameters evaluated were normally distributed, presented homoscedasticity and were subjected to statistical analysis (ANOVA). The information has been added

L243-247 The format of this part is different. Why is it bold?

Thanks for the suggestion. It's a mistake. The text has been corrected and is no longer in bold

L243-247 This sentence is not easy to understand.

Reply: Thanks to the reviewer for the suggestion. The paragraph has been changed. Done

L247 Table 184?

Thanks for the suggestion. It's a mistake. The clone USA 184-411 was meant. The detail has been corrected

Table 2 needs to be reorganized or separated into two tables. The meaning of F, df and P( which should be small p) needs to be described based on the test used.

Dear reviewer. Thank you for your suggestion. The table has been reorganized so that the values only appear on one line. Also, the letter P has been made small.

L255 reference needed for the DMS test and p is small.

DMS has been corrected to LSD (Least Significance Difference), and the reference has been added. The “p” has been changed to small. Details have been added

L265 which statistics was used?

Reply: A generalized lineal model, with a ANOVA (one way) was used. The program used was SPSS, version 21 (IBM SPSS Statistics). Details have been added in section 2.7. Done

Figure 1: Why is hexane included in the statistics? If hexane has a standard deviation of 0 and all other compounds non-zero the homogeneity of variance is surely not true.

Reply: The hexane is the reference because all the substances a diluted in hexane and all the data are relatives (%).

L282 tentatively identified

Thanks for the suggestion. The expressión has been added

L283 all latin names should be in italics

Thanks for the suggestion. But "Salicacea" refers to a family of insects and should not appear in italics

L286 Very vague statement: what is the nature of the compounds that suggests it exactly?

Thanks for the suggestion. The paragraph has been changed. Done

L316 low molecular weight alcohol is a very loosely defined group.  This is the first point where the reason why ethanol is used is specified, this should have been described in the introduction

Reply: Done.

L341 which is the third plot?

Reply: Thanks for the suggestion. It's a mistake. The 2023 results have not been presented because we have run out of funds to travel to the poplar trials during the last few months.

Round 2

Reviewer 1 Report

Comments and Suggestions for Authors

This manuscript has improved from the original submission. In particular, the authors now include a description of the statistical analysis they used.  However, further clarification is needed on exact form of Fisher’s LSD used (see below).  The authors failed to respond to one of the comments I made on their original submission.  They also made some fundamental changes in the description of how the work was done without providing any explanation why and this makes me wonder which version is accurate.

Comment on the original version that were not addressed:

Had any traps been disturbed or damaged or found on the ground during weekly trap collections? If so, how did you deal with the missing data?

Statistics. Did you use the standard Fisher’s Least Significant Difference test or the modified Fisher-Hayter procedure?  The former does not control the experiment-wise alpha at P = 0.05 so the chance of concluding a significant difference between means when it is not true is greater than 5%.  If you used the original LSD test, your conclusions on means comparisons for both antennal response to compounds (Fig. 1) and beetle catch in baited traps (Table 2) may be incorrect. This would be especially true for the EAGs in which you contrasted antennal response to 9 different compounds. 

For additional clarity, below is an excerpt from a chapter in Encyclopedia of Research Design 2010 entitled Fisher’s Least Significant Difference (LSD) Test by Lynne J Williams and Hervé Abdi:

“Note that LSD has more power compared to other post-hoc comparison methods (e.g., the honestly significant difference test, or Tukey test) because the α level for each comparison is not corrected for multiple comparisons. And, because LSD does not correct for multiple comparisons, it severely inflates Type I error (i.e., finding a difference when it does not actually exist). As a consequence, a revised version of the LSD test has been proposed by Hayter (and is known as the Fisher-Hayter procedure) where the modified LSD (MLSD) is used instead of the LSD. The MLSD is computed using the Studentized range distribution q as MLSD = qα,A−1 MSS(A) S . (5) where qα,A−1 is the α level critical value of the Studentized range distribution for a range of A − 1 and for ν = N −A degrees of freedom. The MLSD procedure is more conservative than the LSD, but more powerful than the Tukey approach because the critical value for the Tukey approach is obtained from a Studentized range distribution equal to A. This difference in range makes Tukey’s critical value always larger than the one used for the MLSD and therefore it makes Tukey’s approach more conservative.”

I do not use SPSS so cannot determine if the authors used the original or modified LSD test, but if they used the original LSD, the significant differences in both the EAG and field tests may be inflated, and the data should br reanalyzed using a method that controls the experiment-wise error rate, like Tukey’s test.

Unexplained changes in stated methodology between the original and revised submission:

Line 216  “Four replicates per trap-attractant combination were conducted..”   In your original submission you stated "Two replicates per trap-attractant combination were conducted....". Now you state there were four replicates. I assume the first version was incorrect? But I wonder how such a basic error could be made? Please explain.

Line 218  “Trap position was changed every two weeks in rotation.”  In your original submission you state: "The position of the trap-attractant was not rotated during the months of evaluation." This is another unexplained change in the methods that requires an explanation.  Which version can I assume is accurate?

I have made some additional comments and suggested edits in an annotated pdf.

Author Response

We thank the reviewer for his careful reading and helpful comments, which we have used to improve our manuscript. Please find below a point-by-point list of the answers to your comments.

Comments and Suggestions for Authors

This manuscript has improved from the original submission. In particular, the authors now include a description of the statistical analysis they used.  However, further clarification is needed on exact form of Fisher’s LSD used (see below).  The authors failed to respond to one of the comments I made on their original submission.  They also made some fundamental changes in the description of how the work was done without providing any explanation why and this makes me wonder which version is accurate.

Dear reviewer, we apologize if there were any unanswered questions in the previous version.

All questions have now been answered, the statistical analysis has been modified according to your recommendations, and correct details of the methodologies followed in the field have been given (there were some statements on the number of repetitions or on the rotation of traps, which were mistakenly misspelled, now corrected).

Comment on the original version that were not addressed:

Had any traps been disturbed or damaged or found on the ground during weekly trap collections? If so, how did you deal with the missing data?

Dear reviewer. None of the traps were disturbed or damaged during the field trials.

Statistics. Did you use the standard Fisher’s Least Significant Difference test or the modified Fisher-Hayter procedure?  The former does not control the experiment-wise alpha at P = 0.05 so the chance of concluding a significant difference between means when it is not true is greater than 5%.  If you used the original LSD test, your conclusions on means comparisons for both antennal response to compounds (Fig. 1) and beetle catch in baited traps (Table 2) may be incorrect. This would be especially true for the EAGs in which you contrasted antennal response to 9 different compounds.

Dear reviewer, ANOVA with Fisher's LSD post hoc test was performed on both field captures and insect antennal response. We fully understand your explanation, and therefore we have modified the analysis performed (from DMS to Tuckey) on both data sets.  

For additional clarity, below is an excerpt from a chapter in Encyclopedia of Research Design 2010 entitled Fisher’s Least Significant Difference (LSD) Test by Lynne J Williams and Hervé Abdi:

“Note that LSD has more power compared to other post-hoc comparison methods (e.g., the honestly significant difference test, or Tukey test) because the α level for each comparison is not corrected for multiple comparisons. And, because LSD does not correct for multiple comparisons, it severely inflates Type I error (i.e., finding a difference when it does not actually exist). As a consequence, a revised version of the LSD test has been proposed by Hayter (and is known as the Fisher-Hayter procedure) where the modified LSD (MLSD) is used instead of the LSD. The MLSD is computed using the Studentized range distribution q as MLSD = qα,A−1 MSS(A) S . (5) where qα,A−1 is the α level critical value of the Studentized range distribution for a range of A − 1 and for ν = N −A degrees of freedom. The MLSD procedure is more conservative than the LSD, but more powerful than the Tukey approach because the critical value for the Tukey approach is obtained from a Studentized range distribution equal to A. This difference in range makes Tukey’s critical value always larger than the one used for the MLSD and therefore it makes Tukey’s approach more conservative.”

I do not use SPSS so cannot determine if the authors used the original or modified LSD test, but if they used the original LSD, the significant differences in both the EAG and field tests may be inflated, and the data should br reanalyzed using a method that controls the experiment-wise error rate, like Tukey’s test.

Dear reviewer, thank you very much for your time and dedication in explaining all the problems that can be caused by doing an original or modified LSD analysis.

To avoid all those problems with significance, the data have been analysed with an ANOVA, with a subsequent Tuckey post hoc analysis.

Unexplained changes in stated methodology between the original and revised submission:

Line 216  “Four replicates per trap-attractant combination were conducted..”   In your original submission you stated "Two replicates per trap-attractant combination were conducted....". Now you state there were four replicates. I assume the first version was incorrect? But I wonder how such a basic error could be made? Please explain.

Dear reviewer, I understand your confusion, but as you rightly say, the first version was incorrect, and this new version is the correct one.

"The traps were divided into two blocks (one in the I-214 clone and one in the USA 184-411 clone), each with area of ​​0.4 ha (40 m length × 20 m width). One block contained three trap–attractant combinations (trap-Ethanol, trap-Ethanol+MB and trap-Ethanol+SA). Four replicates per trap–attractant combination were conducted, thus totaling 24 traps in each poplar plantation".

Now appears explained on page 6, from line 297 to line 301.

Line 218  “Trap position was changed every two weeks in rotation.”  In your original submission you state: "The position of the trap-attractant was not rotated during the months of evaluation." This is another unexplained change in the methods that requires an explanation.  Which version can I assume is accurate?

Dear reviewer, again, I understand your confusion. The second version is the correct one. Erroneamente anotamos en la primera versión que las trampas no se rotaron.

All the traps inside each block were randomly distributed. Trap position was changed every two weeks in rotation. All traps were visited once a week (from May to September).

Now appears explained on page 6, lines 301-302.

I have made some additional comments and suggested edits in an annotated pdf.

Dear reviewer, I am sorry to inform you that there are no annotated pdf files on the platform, so I cannot respond to your additional comments.

Reviewer 4 Report

Comments and Suggestions for Authors

Dear  Editor and Authors,

The new manuscript is significantly modified, but many major concerns are still not met and there are serious concerns about the reproducibility and reliability of the results. Thus I cannot support the publication of the manuscript and I recommend rejecting it at this stage.

Contrary to the response, several changes are not actually done in the new version. Additionally there are multiple major and minor comments not even addressed in the response letter.

Even if I understand the publication pressure, if the taxonomical identification is ungoing and the manuscript should be in preprint status until the information can be incorporated? This can have major implications about the interpretation of your research data that the reader of your research will have no chance to learn about.

The also authors admit, many factors affect the volatile profile of the plants under field conditions, not just the genotype. Even if  the SPME volatile collections were done in triplicate, the standard error of relative abundance of the independent biological replicates is not shown (contrary to the response letter that states the section is changed).

Additionally, I am puzzled why no standard devations and means are shown, if there are truly  three volatile collections of independent biological replicates analysed?

Without any of this information it is improper to believe that the differences between the volatile profiles are solely a result of the plant genotypes and not environmental conditions. Showing data distribution and using statistics is not a matter of this field, but a general principle of research design.

The term „ quantitatively different” is still in the text which is because of all these methodological issues completely misleading.  In the letter, the authors claimed that they removed it.

Contaminations from the SPME fiber if the operating temperature is not exceeded (as now described in version 2) will indeed be mostly siloxanes, but blank samples of the containers used for volatile sampling should still be included in the experimental protocol jut as the methods of cleaning volatile collection devices. Without blank samples, carryover of volatile components between samples can be overlooked. Including negative controls is not specific to volatile analysis but a general principle of research design.

Additionally, are the authors sure which SPME fibers they used? Or just deleted one of the two types from the methods? Now PDMS/DVB fibers are described in the methods:

The fibers were 1 cm long and consisted of fused silica support and a 65 μm thick PDMS/DVB (Polydime-thylsiloxane/Divinylbenzene) coating”

but in response to minor comment for L139 they say they used PDMS:

“At the beginning two fibers were tested, but finally just the fiber were 1 cm long and consisted of fused silica supports and 100 μm thick PDMS (Polydimethylsiloxane) coating. We have rewritten this section accordingly.”

The cited reference (Adam R. 2007) does not validate the use of one column. Even those authors in the cited manuscript used multiple columns, but compared to the volatile collection methods it is less of a crutial issue. The general consensus of the field of chemical ecology and analytical chemistry is using two columns. The use of one column is not the new standard of the field due to some kind of improved technology.

If one column is used and retention indexes are calculated the authors need to present them next to the retention indexes found in literature (NIST, Wiley) otherwise the reader cannot evaluate the quality of these matches. The authors stated that they have done the requested change in the table but in fact did not.

It is good that the ANOVA is described, but still hard to believe that the homogenity of variance is met and tested. On Figure 1. hexane has literally no variance (0 standard deviation as it is considered as 100%) all other groups have a big error bars, all bars have letters on top, that shows they were included in the statistical tests. This is another serious conundrum.

Many minor comments could be listed but would divert the focus from these still major issues.

Author Response

We thank the reviewer for his careful reading and helpful comments, which we have used to improve our manuscript. Please find below a point-by-point list of the answers to your comments.

Reviewer 4

Even if I understand the publication pressure, if the taxonomical identification is ungoing and the manuscript should be in preprint status until the information can be incorporated? This can have major implications about the interpretation of your research data that the reader of your research will have no chance to learn about.

Reply: Fortunately, on Monday, November 4 at 6:45 p.m., we received an email from the Scolytidae taxonomists Miloš Knížek, Ph.D. from Forestry and Game Management Research Institute (Czechia) and Åke Lindelöw from Sweden. They found all the specimens sending by us Trypophloeus belonging to the same species and they were identified asTrypophloeus binodulus Ratzeburg, 1837. The name was stated according to the most recent systematics used in Palearctic Catalogue. Keys used are Pfeffer 1995 and Hansen 1956. Done

The also authors admit, many factors affect the volatile profile of the plants under field conditions, not just the genotype. Even if  the SPME volatile collections were done in triplicate, the standard error of relative abundance of the independent biological replicates is not shown (contrary to the response letter that states the section is changed). Additionally, I am puzzled why no standard devations and means are shown, if there are truly  three volatile collections of independent biological replicates analysed?Without any of this information it is improper to believe that the differences between the volatile profiles are solely a result of the plant genotypes and not environmental conditions. Showing data distribution and using statistics is not a matter of this field, but a general principle of research design.

Reply. Thanks for your suggestion! The standard deviations and mean values of all replicates have been added to Table 1. Done.

The term ”quantitatively different” is still in the text which is because of all these methodological issues completely misleading.  In the letter, the authors claimed that they removed it

Reply. Thanks for your suggestion! It does not appear in the new version. Done.

Contaminations from the SPME fiber if the operating temperature is not exceeded (as now described in version 2) will indeed be mostly siloxanes, but blank samples of the containers used for volatile sampling should still be included in the experimental protocol jut as the methods of cleaning volatile collection devices. Without blank samples, carryover of volatile components between samples can be overlooked. Including negative controls is not specific to volatile analysis but a general principle of research design. Additionally, are the authors sure which SPME fibers they used? Or just deleted one of the two types from the methods? Now PDMS/DVB fibers are described in the methods:.....”

Reply: PDMS/DVB (Polydime-thylsiloxane/Divinylbenzene) coating were used. Of course, that all flasks were washed, rinsed with hexane and dried prior to each collection of volatiles is a routine that we had omitted in our previous materials and methods section. It is now included in the new version of the manuscript. Done

The cited reference (Adam R. 2007) does not validate the use of one column. Even those authors in the cited manuscript used multiple columns, but compared to the volatile collection methods it is less of a crutial issue. The general consensus of the field of chemical ecology and analytical chemistry is using two columns. The use of one column is not the new standard of the field due to some kind of improved technology. If one column is used and retention indexes are calculated the authors need to present them next to the retention indexes found in literature (NIST, Wiley) otherwise the reader cannot evaluate the quality of these matches. The authors stated that they have done the requested change in the table but in fact did not......

Reply: More than 60% of MS chromatogram component peak structures were corroborated with the MS of authentic standards. The uncertainty are the hydrocarbon sesquiterpenes, whose standards are not commercial or those available are of such low purity that they are unfeasible as GC standards. The identification of structurally complex natural mixtures of sesquiterpenes from plants is often reported as “tentative”, as authentic standards are not commercially available for most of the sesquiterpenes. Plausible interpretations were achieved with many unsaturated sesquiterpenes by taking retro Diels-Alder cracking, allylic fission with or without involving rearrangement as well as mobility of double bonds. But, our objective is not really to determine the chemical volatile profile of a matrix, but to determine its potential semiochemical activity and its usefulness in controlling the populations of a pest that is endangering poplar plantations. Even papers in chromatography journals, whose objective is the quantification of these sesquiterpenes or the volatile chemical profile of certain matrices, use a single column in the gas chromatograph:  

Cincotta 2015. 410-421; https://doi.org/10.3390/chromatography2030410

Han 2023; DOI: 10.3390/foods12132484

Alotaibi 2023; DOI: 10.3390/insects14070589

Agelopoulos 1998; DOI: 10.1023/A:1022442818196

It is good that the ANOVA is described, but still hard to believe that the homogenity of variance is met and tested. On Figure 1. hexane has literally no variance (0 standard deviation as it is considered as 100%) all other groups have a big error bars, all bars have letters on top, that shows they were included in the statistical tests. This is another serious conundrum.

Reply. Reply. Thanks for your suggestion. 1 µL of hexane (control stimulus) was presented at the beginning of the experiment and after each group of group test compounds. So you must include it in statistical test. Similar studies have also included the stimulus in their statistical analyses. The new version is reworked using Tukey's test. Done

https://doi.org/10.3390/insects10090274

Many minor comments could be listed but would divert the focus from these still major issues.

Round 3

Reviewer 1 Report

Comments and Suggestions for Authors

I thank the authors for following my recommendations re: statistics and apologize for failing to attach the annotated pdf with my last review. I have just two comments:

1. lines 257-257  "Analysis of the headspace of uninfested and infested apple 256 seedlings indicated qualitative differences in odour profiles. "  You have been talking about poplar clones up until now and now you insert a statement about apple seedlings? Either this is a mistake and this sentence needs deleting, or you are citing another study but have forgotten to provide the reference, or you must add more text  to explain why you analyzed the headspace of apple seedlings and show your results. 

2. lines 392-393  "On the one hand, since there is no trap in these tests with an attractant that exclusively contains salicylaldehyde, it must be considered that this has a synergistic effect with ethanol."

Your results do not tell you whether the increased trap catches with SA + ethanol compared to ethanol alone, were an additive effect or a synergistic effect because you do not know the effects of SA by itself.

I have attached an annotated pdf with additional comments and editorial suggestions.

Author Response

Comments to Reviewer 1.

We thank the reviewer for his careful reading and helpful comments, which we have used to improve our manuscript. Our point to point response to yours interesting comments is shown below.

Line 83: “I would say "suggests" rather than "indicates" because host acceptance might occur post-alightment””

Reply: Thanks for your suggestion! That world has been changed by “suggests” Done

Line 133- “prevent evaporation sounds odd when the purpose of the lure is to release ethanol into the atmosphere”

Reply: We would like to explain that, as the bags are zipper-top polybags type they need to be sealed by pressure. The ethanol will be evaporates through the porosity of the polyethylene. Done

Line 256: “apple seedlings? Where did this come from? You have been talking about poplar clones up until now and now you insert a statement about apple seedlings? Either this is a mistake and this sentence needs deleting, or more text must be added to explain why you analyzed the headspace of apple seedlings and the results shown.”

Reply: We thank the reviewer. In the version of the manuscript, the whole sentence has been deleted. We are still working on another study about infested and non infested poplars by T. binodulus and this sentence is outside the scope of this manuscript. Evidently, the word apple was a mistake. Done

 “I said in my first review, you do not need to include uppercase letters to denote a significant difference between clones because there are only two clones. The P value of the ANOVA showing the significance of clone in the right-hand column is alll you need. Please delete the uppercase letters as they only make the Table more cluttered and less clear”

Reply: We fully agree with the Reviewer. This Table has been rewritten and the uppercase letters eliminated in the new version of the manuscript. Done

Reply: Sorry for the mistake. Done

Line 339. “This is rather misleading. The combination of ethanol and alpha-pinene has been used operationally as a general attractant for monitoring of conifer infesting bark beetles and wood borers for decades. Please make it clear in this sentence that you are referring to use of host volatiles from hardwood trees and not host volatiles from all types of plants”

Reply: Thanks for your suggestion! Few details have been added in the new manuscript version. Done

Line 393. “Your results do not tell you whether the increased trap catches with SA + ethanol compared to ethanol alone, were an additive effect or a synergistic effect because you do not know the effects of SA by itself.”

Reply: Thanks for your suggestion! This paragraph has been removed in the new version of the manuscript. Done
